# Phytochemicals and Their Usefulness in the Maintenance of Health

**DOI:** 10.3390/plants13040523

**Published:** 2024-02-15

**Authors:** Elda Victoria Rodríguez-Negrete, Ángel Morales-González, Eduardo Osiris Madrigal-Santillán, Karina Sánchez-Reyes, Isela Álvarez-González, Eduardo Madrigal-Bujaidar, Carmen Valadez-Vega, German Chamorro-Cevallos, Luis Fernando Garcia-Melo, José A. Morales-González

**Affiliations:** 1Servicio de Gastroenterología, Hospital de Especialidades, Centro Médico Nacional Siglo XXI, Mexico City 06720, Mexico; jev_rn@yahoo.com.mx; 2Laboratorio de Medicina de Conservación, Escuela Superior de Medicina, Instituto Politécnico Nacional, Mexico City C.P. 11340, Mexico; eomsmx@yahoo.com.mx; 3Escuela Superior de Cómputo, Instituto Politécnico Nacional, Unidad Profesional ”A. López Mateos”, Ciudad de México 07738, Mexico; 4Servicio de Cirugía General, Hospital de Especialidades, Centro Médico Nacional Siglo XXI, Mexico City 06720, Mexico; drakarinacg@yahoo.com.mx; 5Laboratorio de Genética, Escuela Nacional de Ciencias Biológicas, Instituto Politécnico Nacional, Av. Wilfrido Massieu s/n, Zacatenco, Gustavo A. Madero, Mexico City 07738, Mexico; isela.alvarez@gmail.com (I.Á.-G.); edumadrigal.bujaidar@gmail.com (E.M.-B.); 6Área Académica de Medicina, Instituto de Ciencias de la Salud, Universidad Autónoma del Estado de Hidalgo, Ex-Hacienda de la Concepción, Tilcuautla, San Agustín Tlaxiaca 42080, Mexico; marynavaladez@hotmail.com; 7Laboratorio de Toxicología Preclínica, Departamento de Farmacia, Escuela Nacional de Ciencias Biológicas, Instituto Politécnico Nacional, Mexico City C.P. 07738, Mexico; gchamcev@yahoo.com.mx; 8Laboratorio de Nanotecnología e Ingeniería Molecular, Área Electroquímica, Departamento de Química, CBI, Universidad Autónoma Metropolitana-Iztapalapa, Mexico City 09340, Mexico; electronicfer@hotmail.com

**Keywords:** phytochemicals, health, disease, oxidative stress

## Abstract

Inflammation is the immune system’s first biological response to infection, injury, or irritation. Evidence suggests that the anti-inflammatory effect is mediated by the regulation of various inflammatory cytokines, such as nitric oxide, interleukins, tumor necrosis factor alpha-α, interferon gamma-γ, as well as the non-cytokine mediator, prostaglandin E2. Currently, the mechanism of action and clinical usefulness of phytochemicals is known; their action on the activity of cytokines, free radicals, and oxidative stress. The latter are of great relevance in the development of diseases, such that the evidence collected demonstrates the beneficial effects of phytochemicals in maintaining health. Epidemiological evidence indicates that regular consumption of fruits and vegetables is related to a low risk of developing cancer and other chronic diseases.

## 1. Introduction

Redox homeostasis is a vital process that maintains the equilibrium of radical oxygen species (ROS) and radical nitrogen species (RNS), thus ensuring the normal functioning of the cells. The superoxide radicals (•O_2_), hydrogen peroxide (H_2_O_2_), hydroxyl radical (•OH), and nitric oxide (NO) are the most common free radicals, which are formed as byproducts of cellular metabolism under normal physiological conditions and in response to pathological conditions [1]. The accumulation of ROS gives rise to oxidative stress in the cells, damaging extracellular components such as proteins, lipids, and nucleic acids. In Mexican traditional medicine, the use of plants and plant products goes back several thousand years, when the extracts of some medicinal plants were utilized against diverse types of illnesses. Plants are a rich source of secondary metabolites, commonly denominated phytochemicals; these are grouped as alkaloids, polyphenols, flavonoids, saponins, carotenoids, and terpenes, with diverse structural and functional properties that have been proven in animal and human cells [2,3]. Phytonutrients are natural substances; however, they are not called nutrients in the traditional sense because plants do not synthesize them; instead, they are produced by specific cellular types; they carry out important functions in the secondary metabolism of plants, as insect repellents and sun blockers, as well as growth-regulators, and their effects can be beneficial or damaging to health, depending on the dose [4]. 

Phytochemicals are low-molecular-weight (LMW) secondary metabolites found naturally in plants; they are biologically active molecules that play a significant role in the normal cellular metabolic process and promote health and disease prevention [5,6,7]. The difference between primary and secondary metabolites lies in the fact that the former contributes to the energy metabolism and to the structure of the plant cells, examples of which are proteins, fats, carbohydrates, and dietary fiber; the secondary metabolites comprise non-nutritive dietary components that are essential for the interactions of plants with the environment, and they protect these against insects and fungi (Figure 1). Despite their benefits, phytochemicals can be harmful because they can restrict the availability of nutrients and increase the permeability of the intestinal wall [5]; clinical tests indicate that high doses of antioxidants increase mortality [8]. The exact amount of phytochemicals is unknown; however, it is estimated that this number exceeds 100,000 substances [5]. 

Food composition is important for human health and aging; studies have demonstrated that the amount and quality of nutrients we consume daily are fundamental for changing the conditions of health and disease [9]. In a typical human diet, 1.5 g of phytochemicals is ingested each day, while vegans and vegetarians can exhibit much greater ingestion of secondary metabolites; notwithstanding this, there are no recommended ingestion levels for phytochemicals [4]. 

The classification of phytochemicals is based on their chemical structure and their functional characteristics [4], and the phytochemical groups include carotenoids, common in vegetables and fruits, containing 50 of the 700 natural carotenoids important in human nutrition [10]; beta carotene represents up to 30% of total serum carotenoids. The total daily ingestion of carotenoids reaches 6 mg in the Western diet; phytosterols are closely related to cholesterol due to their chemical structure and are found principally in dried fruits, legumes (alfalfa, peas, beans, white Navy beans, chickpeas, lima beans, green beans, lentils, peanuts, and soy), in seeds and oils, in which daily ingestion ranges on average from 100 to 500 mg [11], while their absorption is less (10%) on comparison with that of cholesterol (40%). This low availability of phytosterols is because they are absorbed in the enterocytes and are actively transported toward the intestinal lumen [4]. The saponins are surfactant compounds that form complexes with proteins and lipids; they are abundant in leguminous plants, their average daily ingestion is 15 mg, and they have a low absorption rate, which confers upon them greater intestinal tract activity [4]. The glucosinolates are present in members of the cabbage family, providing radishes, mustard, and broccoli with their typical flavor; their daily ingestion is between 10 and 50 mg, and this can be duplicated in vegetarians and vegans [12]; polyphenols include phenolic acids and flavonoids (quercetin is the most frequent flavonoid) [4]; flavonoids have many properties, which include antioxidant, anti-inflammatory, analgesic, anti-proliferative, anti-cancer, anti-angiogenic, anti-microbial, and anti-viral [13], they inhibit the production of NO, one of the principal mediators of inflammation, as well as of interleukin-1beta (IL-1β), tumor necrosis factor alpha (TNF-α), and prostaglandin E2 (PGE2) [14]; they are present in red wine, dark chocolate, and are found nearly exclusively in the external layer of vegetables (onions, lettuce, tomatoes, asparagus, cabbage, artichokes, and celery) and fruits (apples, grapes, citrus fruits, cranberries, strawberries, and raspberries), tea, and olive oil [15]. Monoterpenes such as menthol (mint), caraway seeds, and citric oil are active substances in herbs and spices, with daily ingestion of up to 200 mg, and they possess a high grade of biodiversity in humans due to their liposoluble capacity. The phytoestrogens have similar effects to the endogenous estrogens; the two main groups include isoflavones and ligands that can act as estrogens and antiestrogens. The isoflavones are nearly exclusively found in soy and its products, and Western diets provide less than 2 mg/day compared to the Asian and vegetarian diets, which can be from 15 to 40 mg/day. The sulfides include all of the organosulfur compounds of plants, such as garlic, with allicin as the main active principle of the latter [4]. 

Phytochemicals include alkaloids, polysaccharides, and terpenoid polyphenols, which possess low availability because, despite humans consuming various grams of phytochemicals in their daily diet, only a fraction of these compounds is absorbed in the circulation [8]. The phenolic compounds play an important beneficial role in health due to their high antioxidant capacity, entertaining as they do the advantage of escaping from upper digestive tract digestion, allowing them to be absorbed into the plasma during digestion [16]. The polysaccharides exert an anti-inflammatory and analgesic effect, which can be related to the diminution of RNS and associated with the increase in the antioxidant enzymes (catalase [CAT], superoxide dismutase [SOD], and glutathione peroxidase [GPx]) [14,17]. For example, only 16% of resveratrol consumed orally is absorbed in the blood and excreted in the urine 24 h after its consumption, while the levels of catechin and quercetin are even lower (2% and 5%, respectively). Ninety percent of the polyphenols in a cup of green tea are eliminated from circulation at 8 h, and this can vary between individuals. The non-absorbed fraction of phytochemicals promotes intestinal functions and acts as a prebiotic, while the absorbed part induces mechanisms of resistance to stress, which improves cellular and organic functions [8]. The flavonoids are metabolized in the small intestine and the liver [15]; the properties that they possess include antioxidant, anti-inflammatory, analgesic, antiproliferative, anticancer, anti-angiogenic, antimicrobial, antiviral, and neuroprotective activity [18]. The soy saponins significantly inhibit the release of PGE_2_, NO, TNF-α, and of the monocyte chemoattractant protein 1 (MCP-1) in a dose-dependent manner and are similarly useful for suppressing the activity of the nuclear factor of the activated kappa beta-cell light chains (NF-κB), increasing the activity of glutathione, SOD, and CAT [19]. 

Inflammation is a reaction of the body to a foreign substance (viral, bacterial, parasitic, or chemical). These substances are recognized by the immune cells through various proinflammatory pathways, leading to the production of cytokines and the activation of the immune cells, such as macrophages and lymphocytes, which will be charged with eliminating the foreign substances. However, if the organism fails in this elimination process, the inflammation increases, generating a chronic phase characterized by the overproduction of cytokines, chemokines, and inflammatory enzymes. The flavonoids, with their anti-inflammatory properties, can interact with many molecules involved in the inflammatory pathways, diminishing the activity of cytokines (NF-κB, IL-1β, TNF-α, IL-6, IL-8, and cyclooxygenase 2 [COX-2]), chemokines, and also of inflammatory enzymes. Catechins and quercetin could increase the production of IL-10, an anti-inflammatory compound, through the combined inhibition of IL-1β and TNF-α; other flavonoids inhibit arachidonic acid, phospholipase A_2_, the COX, and the RNS [15]. 

Rutin, hesperidin, and quercetin reduce chronic and acute inflammation; it has been observed that diosmin and hesperidin reduce the synthesis of leukotriene B4 (LTB_4_), accompanying the improvement of inflammation of the colon. Silymarin is a flavonoid subtype that reduces the hypoxia-inducible factor 1 alpha (HIF-1α) and the induction of nitric acid synthase (iNOS) using the inhibition of NF-κB. The principal component of silymarin, silybinin, diminishes the epidermal growth factor receptor (EGFR), thus reducing hypotrophy. It has been demonstrated that flavonoids increase the activity of the natural killer (NK) cells and the cytotoxic C cells; they induce transcription factors such as nuclear factor erythroid 2-related-factor 2 (Nrf-2) that mediate the expression of the antioxidant proteins. Nrf-2 suppresses the expression of the monocyte MCP-1 and of the vascular cell adhesion molecule 1 (VCAM-1), which diminishes the formation of atherosclerotic lesions in rats and rabbits [15]. 

Flavonoids are exogenous antioxidants; they reduce the reactive species on inhibiting iNOS, xanthine oxide synthase, or regulation of the ionic channels, and they reduce the oxidation of low-density lipoproteins (LDL). Apigenin diminishes the markers of oxidative stress, such as GPx, SOD, and the activation of caspase-3; another of their functions is exerting their antibacterial action on inhibiting nucleic acid synthesis, it diminishes the energy metabolism, and it modifies the function of the cellular membrane. Some flavonoids exhibit cytotoxic and cytostatic activity in tumor cells [15]. Metabolites extracted from the plant material are used to induce apoptosis in cancer cells. The herbal medicines are tested both in vitro and in vivo; the anticancer activities of the various medicinal plants have been tested in vivo using different animal models. Although plant-based compounds have been shown to be less toxic compared to conventional synthetic compounds, there is growing evidence of the side effects of the unregulated use of these plants against different diseases [20]. Flavonoids act at different stages of viral infection, such as viral entry, replication, and protein translation. There are several mechanisms by which flavonoid phytochemicals inhibit and act on the viruses. They can obstruct the attachment and entrance of viruses into the cells and hamper different phases of viral DNA replication, protein translation, and poly-protein processing. They can also inhibit the release of viruses to invade other healthy host cells. Even though their mechanisms of action are not fully clarified, polyphenols seem to damage bacterial cell membranes or interfere with the production of amino acids needed for bacterial growth [21]. Phytochemicals exert their antibacterial activity through different mechanisms of action, such as damage to the bacterial membrane and suppression of virulence factors, including inhibiting the activity of enzymes and toxins and bacterial biofilm formation [22]. Phytochemical investigations exhibited that *Actinidia eriantha* Benth (AE) contains rich phytochemicals, like terpenoids, alcohols, phenolics, aldehydes, organic acids, flavonoids glycosides, ketones, and glucoside. A 24 h administration of an ethyl acetate fraction from the root of *Actinidia eriantha* Benth (AE) EA-EER (100 μg/mL) significantly suppressed cell proliferation by blocking the G1 to S cell cycle progression and downregulating the VEGF-A (vascular endothelial growth factor A) and VEGFR-2 (vascular endothelial growth factor receptor 2) expression [23]. 

There are many methods by which essential oil (EO) exposure can occur, including inhalation, ingestion, massage, and skin application. EOs are known for many of their health effects, such as their antibacterial, antibiotic, and antiviral properties. The oils are typically extracted via steam distillation of plants; most EOs are generally safe, and most adverse effects are mild. Terpenes are the biggest class of chemicals found in EOs. There are several classes of terpenes, but the most important in EO are the monoterpenes and sequiterpenes; the distinct smell is produced from these two groups of chemicals [24]. 

Several EOs have a strong interest in research for their cytotoxic capacity. The main mechanisms that mediate the cytotoxic effects of EO include the induction of cell death by activation of apoptosis and/or necrosis processes, cell cycle arrest, and loss of function of essential organelles [25]. The effect antibacterial activity of EO may be bacteriostatic or bactericidal; the mechanism of antibacterial action is facilitated by a succession of biochemical reactions within the bacterial cell; due to lipophilic compounds, EOs easily penetrate bacterial cell membranes and have been reported to disrupt critical processes of the cell membrane like nutrient processing, synthesis of structural molecules, energy generation, emission of growth regulators, and influences on the cell–cell communication. Some of the EOs commonly used come from garlic, ginger, clove, black pepper, green chile, cinnamon, pimiento, thyme, oregano, and rosemary. The microorganisms mainly inhibited by essential oils are *S. aureus*, *V. parahemolyticus*, *C. perfringens*, *polio virus*, *Adeno virus*, *L. monocytogenes*, *Candida* spp., *Enterobacteriaceae*, *P. mirabilis*, *S. pyogenes*, and *E. coli*. EOs might interfere with virion envelopment, designed for entry into human host cells, synthesis of viral proteins, inhibition of the early gene expression process, glycosylation process of viral, and inhibition of virus replication by hindering cellular DNA polymerase [26]. 

Essential oils, such as chamomile, eucalyptus, rosemary, lavender, and millefolia, have been found to mediate the inflammatory response; they have the ability to influence antioxidant activity, signaling cascades, cytokines, regulatory transcription factors, and the expression of pro-inflammatory genes. The three main anti-inflammation properties of EOs include inhibiting arachidonic metabolism, cytokine production, and pro-inflammatory gene expressions [26]. 

## 2. Function of the Phytochemicals in the Organism 

### 2.1. Intestinal Diseases 

The intestinal microbiota (IM) represents a great composition, nearly 4 × 10^13^, of unique and intricate microbes that reside in the gastrointestinal tract (GI), the majority presenting in the colon and, to a lesser degree, in the upper digestive tract. Thus, it is that in the stomach, due to its acidity, we find therein approximately 10 bacteria/g, 1000 bacteria/g in the duodenum, 10,000 bacteria/g in the jejunum, 10 million bacteria/g in the ileum, and 10^12^ bacteria/g in the colon (Figure 2). The principal bacteria comprise *Bacteroides, Firmicutes*, *Actinobacteria*, *Proteobacteria*, and *Verrucomicrobia*, with the former two representing approximately 90% of the species [27,28].

The IM depends on the host and provides benefits for the same [20] because it synthesizes certain vitamins (vitamin K, biotin, cobalamin, folates, nicotinic acid, pantothenic acid, pyridoxine, riboflavin, and thiamin), which are important for the bacterial metabolism [29]; these can provide energy (around 10% of the required daily energy) through the epithelial absorption of the bacterial products such as short-chain fatty acids (SCFA) (butyrate, propionate, and acetate). The IM plays a critical role in health, preserving the neuroendocrine, metabolic, and immune functions; therefore, dysbiosis (disequilibrium in the diversity and composition of the microbiotas) of the IM has shown a relation with alteration in the intestinal barrier, thus reducing the bacterial diversity, altering the immune response, and increasing the risk of inflammatory diseases (Figure 3). The intestine-associated lymphoid tissue is an integral part of the immune system, protecting the body against invasive microorganisms [27].

Similarly, the role of the intestine–liver axis in non-alcoholic fatty liver disease (NAFLD) has been studied, observing that IM plays a vital role in the metabolism [30]; at the same time, the brain–intestine axis implicated in neurodegenerative disorders, such as multiple sclerosis, is due to the intestinal dysbiosis that is associated with endotoxemia, intestinal, and systemic inflammation, triggering neuroinflammation [31] (Figure 4). The colonic microbiota is a key component of the intestine–brain axis. Due to the production of neurotransmitters in the brain, the intestine interacts bidirectionally through the central nervous system (CNS), the hypothalamic–suprarenal pituitary axis, and the endocrine and immunological axis. The colonic microbiota generates serotonin and γ-aminobutyric acid, both of which modulate emotions and behavior. They affect neural signaling and the digestive system surveillance site of the host; the GI tract is the principal immune-surveillance site; on the other hand, serotonin, on being secreted into the GI tract, modulates peristalsis, inflammation, and the development of the intestinal epithelium [32]. The polyphenols exert action on the intestine–brain axis by presenting neurotransmitter activity after crossing the blood–brain barrier, reducing oxidative stress and presenting anti-inflammatory and antioxidant activity on acting on the NF-κB and Nrf-2 transcription factors, respectively [27]. 

The IM plays an important role in psychological behavior by regulating the integrity of the intestinal barrier and the systemic inflammatory responses; the latter was evaluated in an experiment with mice; the administration of sesame oil, rich in lignans (50 mg/kg/day), produced diminution in depression and anxiety, together with a reduction in the level of inflammation at the CNS level, conditioned by the reduction in the levels of IL-6 and TNF-α, associated with the fact that the lignans improve the properties of the intestinal barrier, diminish the levels of lipopolysaccharides (LPS) in plasma. They increase the levels of *Bacteroides* [33]. 

The microbiome contributes to the homostatic regulation of different tissues within the body; the interrelationship between the human and the microbiota can be considered a mutualistic symbiosis, while eubiosis refers to a healthy equilibrium of the microbiomes in the intestine, and this can be observed as altered, which leads to the development of diverse chronic diseases accompanied by an underlying inflammatory condition [34]. Phytochemicals are actively metabolized and excreted by means of the bile and the urine; they possess a short half-life ranging from minutes to hours; some of these, such as berberine, catechin (found in onions, tea, and apples), and resveratrol enrich the beneficial intestinal bacteria, including *Akkermansia*, *Bifidobacterium*, and *Lactobacillus*, and reduce the levels of opportunistic bacteria, including those of *Escherichia* and *Enterococcus.* The polyphenol substrates and catabolites modulate the composition of the IM; the previously mentioned polyphenols are found in coffee, green/black tea, and red wine; this latter reduces the incidence of *Enterobacter cloacae* in patients suffering from metabolic syndrome [28]. The polysaccharide phytochemicals can favor the growth of beneficial bacteria that produce SCFA, which induces appetite reduction in the host. They diminish insulin resistance, avoid the accumulation of lipids, and diminish inflammation [8]; at the same time, they provide food substrates for colonic bacteria and for the polysaccharide proliferation of bacteria charged with degrading the intestinal mucus to obtain energy, thus protecting the intestinal mucosa from erosion, infection, and inflammation. In the absence of phytochemicals and dietary fiber, the proliferation is favored by bacteria that degrade the intestinal mucosa, this layer compromising the intestinal barrier, leading to the translocation of bacterial LPS to the blood, a condition known as metabolic endotoxemia [34]. 

The three SCFA greatest in abundance in the feces include acetate, propionate, and butyrate; it could be said that the latter of these is the most important for human health, in that it is an energy source for the colonocytes, having as they do antioncogenic activity through their capacity to induce the apoptosis of colon cancer cells. Propionate possesses activity in intestinal gluconeogenesis and is also transferred to the liver, where it plays a role in liver gluconeogenesis. Acetate is the most abundant SCFA, and it is essential for bacterial growth; it is transported to the peripheral tissues and utilized in the metabolism of cholesterol and lipogenesis, in addition to playing an important role in the central regulation of the appetite [29]. The relation is well established of the presence of neurological disorders such as Alzheimer’s disease (AD), Parkinson’s disease, multiple sclerosis, autistic spectrum disorder, and depression [32]. The butyrate-producing bacteria (*Firmicutes*) are considered beneficial because their depletion is associated with the development of type-2 diabetes, inflammatory bowel disease (IBD), and irritable bowel syndrome. Dietary fiber can exert an influence on the IM, maintaining an equilibrium in the health/disease state in that it favors the maintenance of the integrity of the tight junctions between a thick and stable mucosa layer; this allows for the production of SCFA that act as prebiotics and also aiding in the integrity of the barrier function [27]. 

The effect of *Akkermansia muciniphila* on intestinal health has been studied. This Gram-negative bacterium stimulates the growth of the mucosal layer; its presence reduces the presence of bacterial pathogens in this layer; it derives from the consumption of a diet rich in polyphenols, among which we find grapes and cranberries [35]. In a study conducted in rats, these were administered at 1 mg of resveratrol/kg/day for 25 days, which demonstrated an increase in *Lactobacilos* spp. and *Bifidobacterium* spp. and a reduction in *Enterobacterias* spp. Likewise, the extract of red wine, rich in proanthocyanidins, revealed an improvement in the species of *Bacteroides* and *Lactobacilos* and, at the same time, reduced those of *Clostridium* spp. The flavone naringenin inhibits the growth and adhesion of *Salmonella typhimurium*, the causal bacterium of diarrhea [27]. The carotenoids favor the production of immunoglobin A (IgA), preventing intestinal dysbiosis; capsaicin is a carotenoid found in the red chili pepper; on being administered at a dose of 2 mg/kg during 12 weeks, it demonstrated a reduction in weight and a diminution in blood lipids as well as of glucose. Another studied carotenoid is astaxanthin, present in algae and shellfish; this was evaluated in the mouse model of mice with NAFLD administered with 50 mg of astaxanthin/day during 12 weeks, significantly reducing the accumulation of lipids and the serum markers of liver damage increasing the number of beneficial bacteria [36]. 

The colonic microbiota is capable of modifying the structure and properties of the bile acids, the latter possessing antibacterial characteristics, exercising a detergent effect on the bacterial cellular membranes, and possessing the capacity to induce damage to the DNA and to alter the protein structures [29], due to the latter, the bile acids appear to play a decisive role in the homeostasis of the IM, which has been demonstrated in that their elevated levels are associated with the presence of chronic diseases such as cancer of the liver and IBD [37]. 

While some phytochemicals can activate cellular processes, such as the expression of the detoxification of the antioxidant enzymes, the majority of phytochemicals generate mild stress, which improves the function of the organs. The concept of hormesis has been utilized to explain the beneficial effects of mild stress on the human body. Although stress at high doses produces toxic effects, it is beneficial at low doses because it improves cellular function by inducing mechanisms of resistance and recovery; the same occurs with exercise, in that the latter produces benefits for health when it is engaged in intermittently, with rest periods necessary for developing an adaptive and protector response [38]. In the same manner, physical training is a potent stimulus for promoting the strengthening of cellular defenses [39], in turn supporting that older and sedentary persons can restore their Nrf-2 signaling and an antioxidant response with the practice of physical training [40]. 

Piperin is an alkaloid that possesses antimicrobial activities; at high doses of 3–100 mg/mL, it exerts activity against *Candida albicans*, while at a low dose of 0.1–0.6 mg/mL, it is active against *Pseudomona aeruginosa* [27]; other effects that have been described include the following: antiseptic, improvement of digestion, and an antibacterial and insecticidal effect; however, if consumed in large amounts, it can have negative effects on health [41]. 

Aging is associated with the deterioration of multiple bodily functions and the disease course of inflammation, which leads to the appearance of fragility. The fragility that accompanies aging implies the failures of multiple physiological systems and a persistent activation of the innate immune inflammatory response, which has been related to changes in the IM that are favored by a restricted diet during aging. The Mediterranean diet is characterized by increasing the consumption of vegetables, legumes, fruits, nuts, olive oil, and fish and low consumption of red meat, dairy products, and saturated fats; this diet increases antioxidant activity, generates beneficial changes in the IM, and reduces the incidence of diseases because it improves the levels of anti-inflammatory cytokines (IL-10) and with this, the risk of fragility [42]; contrariwise, a diet rich in fats conditions the inflammation of the mucosa, generating H_2_O_2_ and increasing the proliferation of *Escherichia coli*, in this manner modifying the microbiota [43]. 

### 2.2. Inflammatory Bowel Disease (IBD) 

IBD represents a group of intestinal disorders that gives rise to prolonged digestive-tract inflammation and that presents as a result of interactions among environmental, microbial, and immune factors in a genetically susceptible host; compromised intestinal permeability is one of the principal causal factors of IBD, including Crohn disease and ulcerative colitis [44]. The flavonoids demonstrate a beneficial effect on reducing the inflammatory process [45]. The nucleotide 2 oligomerization domain (NOD2) is expressed in the cells of the intestinal epithelium; it functions as a defensive factor against intracellular bacteria and contributes to the immune response of the guest microbia. It has been shown in rats that the deficiency of NOD2 conditions an altered microbiome, aiding in the presence of colitis and, in addition to that, it conditions the bacteria that are hosts, such as *Bacteroides vulgatus*, which has been related to alterations in the function of the mucosal barrier. The mutation in humans of NOD2 conditions a diminution in the levels of IL-10, an anti-inflammatory cytokine, increasing the number of bacteria of the species *Escherichia*. The caspase recruitment protein 9 domain (CARD 9) is equally required to produce inflammatory cytokines in response to specific bacterial or viral stimuli. There is evidence suggesting that an episode of infectious gastroenteritis can trigger an abnormal immune response with a susceptible IM profile, increasing by 40% the risk of a later development of IBD [45]. 

The oligosaccharides of maternal milk exert prebiotic effects that contribute to the establishment of the IM in the lactating infant, permitting a greater abundance of bacteria such as *Firmicutes* and *Actinobacterias* in comparison to formula-fed children [46]. A diet high in fiber and the consumption of fruit diminishes the risk of IBD; on the other hand, tobacco consumption reduces the beneficial species of *Bifidocaterias* [45]. 

Resveratrol is obtained from the grape and is extracted during the elaboration of red wine; it is known for its antioxidant, anti-inflammatory, and antiproliferative effects; it improves the inflammatory conditions in IBD at the mucosal level and is absorbed by the erythrocytes; however, its concentration in plasma is low due to a high rate of intestinal and liver metabolism. Patients with IBD exhibit elevated levels of reactive species; consequently, high oxidative stress levels conditioned mucosal damage [47]. Mitochondrial dysfunction, in turn, acts on the pathogenesis of IBD because of oxidative stress and the altered production of ATP, which are key points in the development and progression of the disease; it has been demonstrated that resveratrol importantly inhibits mitochondrial dysfunction and prevents the initiation of programmed cell death [48]; it is also capable of protecting against the GI toxicity of chemotherapeutic agents such as methotrexate, demonstrating that it can reduce oxidative stress and induce an antioxidant response in duodenal and jejunal tissues [47]. 

NF-κB is an important regulator of immunity and actively participates in the progression of IBD. Once activated, the transcription factor causes the induction of genes implicated in the inflammatory process and immunity, the molecules of adhesion, and enzymes such as inducible iNO and COX-2. The induction of the expression of cytokines via NF-κB is responsible for the stimulation, activation, and differentiation of the immune cells of the lamina propia, which results in persistent mucosal inflammation [49]. With its antioxidant and anticancer activity, Ellagic acid reduces the expression of IL-6 of COX-2 by 78%, suggesting that it is useful in the diminution of oxidative stress and in the reduction in intestinal permeability. On the other hand, curcumin and allicin reduce the histological scale of the colon and lessen the activity of the colonic myeloperoxidase (MPO), an important marker of oxidative stress; the latter prevents damage to the mucosa of the intestinal epithelial layer, in that it reduces the infiltration of NF-κB. The terpenoids, alkaloids, and quinones diminish the inflammatory process by favoring the anti-inflammatory cytokines (IL-4, IL-10, IL-11, and IL-13) and improving intestinal permeability [44]. *Pulsatilla chinensis* saponins have many pharmacological activities, including anti-tumor, anti-inflammatory, anti-oxidation, anti-virus, anti-schistosome, immune enhancement, and other pharmacological activities. *Pulsatilla* significantly reduced the mesenteric blood flow in UC rats and significantly alleviated the inflammatory response, which indicates that saponins are involved in the anti-UC effects of *Pulsatilla chinensis* [50]. 

### 2.3. Liver Diseases 

The liver is the organ responsible for metabolizing drugs and toxic chemical substances; therefore, it is the main organ objective of diverse exogenic toxins [51]. The principal etiologies of liver diseases are the abuse of alcohol, infection by the hepatitis virus, and metabolic syndrome. The pathological processes that condition liver damage include oxidative stress, lipid peroxidation, inflammation, and the alteration of the immune response. In contrast, others that are mentioned include cytochrome P450 dysfunction, as well as mitochondrial dysfunction, with chronic liver inflammation promoting the progression of liver diseases [52]. Polyphenols have been employed in the treatment of liver disease, the latter based on the underlying mechanisms implicated principally in improving antioxidant defense enzymes through the mediation of the expression of Nrf-2/cytochrome P450 2E1 (CYP2E1), which improves the inflammatory process. Puerarin is obtained from the plant *Pueraria montana* var. *lobata*, a subspecies of *Pueraria montana*, a natural flavonoid that reduces the production of ROS, renovates the antioxidant enzymatic system, regulates the expression of the *S* genes, biosynthesis, and lipid metabolism in liver; it is used in soups, creams, stews, fillings, sauces, or jams [52]. Curcumin significantly attenuates the liver failure caused by LPS, diminishes the serum levels of hepatic transaminases (AST, ALT) and alkaline phosphatase (ALP), and improves the levels of antioxidant enzymes [53]; likewise, silymarin, *Silybum marianum*, has exhibited its usefulness in diminishing liver damage and the inflammatory process [52]. 

Alcoholic liver disease (ALD) is the direct consequence of the metabolism of ethanol, which conditions the products of ROS, mitochondrial damage, and liver steatosis, which are the common characteristics of chronic and acute exposure to alcohol. The benefit of the treatment with polyphenols has been related to the regulation of lipid metabolism and the antioxidant effect [54]. The liver burden of iron has been evidenced as a pathogenic factor in ALD, a cachetin that has demonstrated the diminution of that damage by means of its chelating function by inhibiting the liver absorption of iron and also inhibiting the absorption of iron in the small intestine, thus reducing the serum and liver levels of iron [52]. 

NAFLD is the most common cause of liver disease at the worldwide level. It is defined as the increase in the accumulation of intrahepatic fat in the absence of risk factors such as, for example, alcohol abuse, drugs that favor the production of steatosis, and other causes of chronic liver disease [55]; NAFLD can lead to the development of non-alcoholic steatohepatitis (NASH) in 10–20% of patients, a severe form of fat-induced liver damage conditioned by a disruption in the metabolism of cholesterol, triggering the accumulation of lipids in the liver and with this inflammation, therefore, permitting the progression of liver insufficiency due to the development of fibrosis and cirrhosis [56]. The body mass index (BMI) has a significant correlation with the increase in TNF-α and C reactive protein; oxidative stress and lipid peroxidation comprise the main components that favor the development of cirrhosis and hepatocellular carcinoma [55]. Various phytochemicals possess beneficial effects due to the diminution of circulating cholesterol levels and the prevention of lipid oxidation [56]. Type-2 diabetes is considered an independent risk factor for the development and progression of severe fibrosis in patients with fatty liver, having an annual rate of 0.09% (95% CI 0.06–0.12). The progression of steatosis to steatohepatitis and cirrhosis is closely related to oxidative stress, lipotoxicity, and local inflammation. Glutathione S transferase is the principal intracellular antioxidant and is essential for liver function; a study of knock-out mice observed that the loss in GSH in the hepatocytes leads to steatosis at early ages with mitochondrial damage and liver insufficiency, with the most important transcription factor in the regulation of antioxidant enzymes being Nrf-2 [57]. Patients with underlying liver disease are predisposed to drug-induced liver injury (DILI); patients with NAFLD exhibit an increase in the activity of the CYP2E1, a 1.9–3.1-times diminution in the activity of CYP3A4 in NAFLD, and of NASH, respectively. The antioxidant properties of various natural compounds related to Nrf-2 activity have been investigated; these studies mainly focused on the control of glucose in the blood, with this preventing or delaying the complications due to diabetes; however, the results are contradictory; therefore, sufficient clinical evidence to date is lacking to justify the use of Nrf-2 in the practice of medicine [58]. 

The oxidative stress in NAFLD can induce alterations in the endothelial function occasioned by the formation and deposit of oxidized LDL; it has been determined that the administration of curcumin improves the histopathologic changes of NAFLD on reducing the levels of steatosis and that curcumin normalizes the numbers of serum aminotransferases because the treatment with curcumin [59] reduces oxidative stress, proinflammatory cytokines, IFN-γ, and IL-1β; on the other hand, it reduces the possibility of the progression of NASH to hepatocellular carcinoma [60]. 

Various flavonoids, one of these extracted from green tea, have been useful for impeding the entry of the hepatitis B virus (HBV) and the hepatitis C virus (HCV); silybin is a direct inhibitor of the ARN-dependent polymerase of the HCV. The epicatechins, also composed of green tea, can hinder the replication of the HCV using the inhibition of the protein NS5B and through COX-2, in addition to improving the inflammation induced by the virus; epigenin and quercetin significantly reduce the replication of the viral genome, the latter via the inhibition of the activity of protease NS3 [46]. *Equinacea* (50 mg/kg) and chichoric acid significantly reduce the viral burden of HBV and suppress the expression of the surface antigen and the HBV [61]. 

Any cause of liver damage can lead to the presence of liver fibrosis characterized by the accumulation of collagen and of the extracellular matrix protein (ECM), conditioning the substitution of the normal liver parenchyma via fibrosis, with the hepatic stellate cells (HSC) being those mainly responsible for the deposit of the ECM [62,63]. Polyphenols such as morin, derived from quercetin, gallic acid, and curcumin, exercise an anti-fiber effect on promoting the apoptosis of the ECM, and they attenuate the oxidative stress associated with NF-κB and TNF-α signaling [63]. In a blinded, double-randomized, placebo-controlled study, a 2.150 mg dose of resveratrol capsules was administered twice daily for 3 months, observing a significant reduction in the levels of TNF-α and improvement in the levels of adiponectin in patients with NAFLD [63]. In another study, the usefulness was evaluated of curcumin at a dose of 1000 mg/day for 8 weeks compared with the placebo, determining that curcumin favors the reduction of the BMI, improves liver findings using echography, and reduces the serum levels of aminotransferases [64]. Nuciferine is an alkaloid with an important effect on reducing intracellular triglyceride levels and improving lipid metabolism; supplementation of this compound significantly improves serum and hepatic lipids [65]. It has also been shown to prevent hepatic steatosis in golden hamsters fed with a high-fat diet (HFD). Supplementation with this compound significantly alleviated serum and hepatic lipid levels and enhanced expression levels of PPARα and carnitine palmitoyl transferase 1 (CPT1), which contributed to increased fatty acid oxidation. In addition, nuciferine treatment significantly reversed the increase in hepatic levels of proteins SREBP-1c and fatty acid synthase caused by the HFD, indicating that nuciferine can suppress the development of hepatic steatosis induced by an HFD [66].

The polyunsaturated omega 3 fats regulate the transcription factors associated with the metabolism of lipids in the liver, resulting in greater oxidation of the fatty acids and, with that, the regulation of proinflammatory genes [67]; they reduce the circulating triglycerides by reducing the liver secretion of the VLDL, of the cholesterol, or they increase the metabolism of chylomicrons. The carotenoids are soluble pigments in lipids; they are divided into carotenes and xantophylls; the administration of astaxanthin, pertaining to the group of xantophylls, removes the peripheral-tissue cholesterol to the liver, resulting in the improvement of the serum lipid profile with a reduction in the levels of triglycerides (38%), cholesterol (33%), and LDL (32%). The polyphenols and the phytosterols protect the liver from the damage caused by NAFLD. *Sinphonaris compressa* gives rise to the diminution of 18% of the total cholesterol; there is a reduction in steatosis in the histology of the liver, which lessens the damage to the hepatic parenchyma [56]. Fungi exert a potential impact on human health; the bioactive compounds contain anti-inflammatory, anticancer, antioxidant, and anticholesterolemic properties. *Pleurotus ostreatus* is a polysaccharide with the capacity to produce lovastatin, an agent that aids in diminishing cholesterol levels by means of the inhibition of hydroxymethylglutaryl-CoA reductase, generating a reduction in the triglyceride and cholesterol levels [68]. *Astragalus*, a derivative of *Pleurotus ostreatus*, inhibits the growth of hepatocellular carcinoma because it exerts an influence on the immunoregulatory properties involved in tumor activity [69]; the same effect has been observed when combining curcumin with piperin [70]. *Suillus luteus* and *Boletus bodius* are edible fungi rich in polyphenols; the polyphenols, as well as the dietary fiber of the fungi, can reduce the cholesterol and the bile acids produced at the liver level; the bile acids are secreted in the small intestine in order to facilitate the digestion and absorption of the fats in the diet, which are reabsorbed by the erythrocytes and transported in return to the liver through the enterohepatic circulation, and the bile acids are secreted, together with the cholesterol, through the feces [68]. Panax notoginseng saponins (PNS) exhibited potent anti-lipogenesis and anti-fibrotic effects in NAFLD mice that were associated with the TLR4-induced inflammatory signaling pathway in the liver. More strikingly, PNS treatment caused a deceleration in the gut-to-liver translocation of microbiota-derived SCFA products [71]. 

Pharmacological experiments with *Equinacea* have confirmed its immunomodulatory, antiviral, antibacterial, anti-inflammatory, antioxidant, and antitumor activity, of hepatoprotection, and cicatrization; however, some studies show its association with hepatopathy. *Equinacea* improves the hepatotoxicity induced by xenobiotics in that it controls oxidative stress and promotes apoptosis. *Equinacea* possesses importance in the treatment of DILI, which can give rise to acute liver failure and death in some cases; for example, the chronic use of methotrexate reduces the antioxidants at the hepatic level and induces oxidative stress; in rats, it has been observed that the administration of an antioxidant such as chichoric acid (25 mg/kg) can improve the histologic alterations and improve the tests for aminotransferases, and ALP; the underlying mechanism favoring this is due to the attenuation of oxidative stress and the inflammatory process [72]. 

Ferroptosis is a form of iron-dependent cellular death that is accompanied by high levels of lipid peroxidation; the liver is the most important organ for the deposit of iron; this deposit has been observed to increase in chronic liver diseases; certain phytochemicals inhibit ferroptosis in order to protect the liver from the development of NASH, DILI, and acute liver failure [73].

Infection by COVID-19, the cytokine storm, is an important source of endogenous oxidative stress; there is excessive production of free radicals and RNS, favoring the increase in cytokines, giving rise to an exaggerated inflammatory response on initiation. The cytokines and the endotoxins stimulate the isoforms of the iNO, increasing the production of NO, resulting in toxicity to the mitochondria and generating mitochondrial dysfunction and cytopathic hypoxia; the host’s response to the stress comprises a marked increase in the levels of cortisol with the purpose of reducing the inflammatory process and of preventing tissue damage [74]. The incidence of liver damage was reported to be between 14% and 53%; approximately one-third of the affected population presented alteration in the levels of this latter damage, which conditions an alteration in the depurification of lactate and modifies the catabolism of proteins. As a consequence of hyperaminoacidemia and hyperammonemia, there is presentation of malnutrition. The phytochemicals induce protection against SARS-CoV-2 due to the cytoprotection system, which is found in the species *Brassica*, such as broccoli, a potent bioactivator of Nrf-2 [75]; silymarin is a mixture of seven flavonoids extracted from *Silybum marianum*; it has been considered as one of the main phytochemicals in action against infection by COVID. The mechanism of action of silymarin is not yet fully elucidated but involves triggering the Nrf-2 (known as a master regulator of the cytoprotective response)/Keap1/ARE (antioxidant response element) signaling pathway [76]; Nrf2 is additionally implicated in oxidative stress induced by alcohol consumption, upregulating antioxidant defense genes and downregulating the genes involved in lipogenesis [77]. Curcumin is another phytochemical with potential for SARS-CoV-2, as well as the liver damage induced by the same, the latter because the administration of curcumin for 14 days induces antioxidant enzymes, among these CAT, SOD, and GPx, inhibiting the free radical-producing enzymes and removing the peroxide radicals, H_2_O_2_, hydroxyl radicals, and NO [78].

### 2.4. Metabolic Syndrome

The excessive accumulation of energy due to the ingestion of a diet high in fat and sugar causes the metabolic syndrome; this is characterized by the elevated expression of TNF-α, an adipokine produced from enlarged adipocytes; it suppresses the production of the type-4 glucose transporter (GLUT 4) and induces insulin resistance, responsible for the chronic inflammation and development of diabetes, and additionally, it reduces the lipase lipoprotein activity in the adipocytes, stimulating hepatic lipolysis [79]. *Firmicutes* are Gram-positive bacteria that predominate in persons with obesity because the *Firmicutes* convert cellulose into glucose, which could contribute to obesity. After the initiation of obesity, the adipocytes constantly produce free fatty acids; the continuous production of these affects the macrophages, causing these to produce TNF-α and inhibiting GLUT 4, finally conditioning the development of diabetes. On the other hand, a diminution of *Bacteroides*, which are Gram-negative bacteria, induces a fine layer of mucin in the colon, which debilitates the tight cellular junctions, giving rise to hypertension and dyslipidemia. Consuming prebiotics, probiotics, polyphenols, and vitamin D is a good strategy for maintaining good health. The phytochemicals of citrus fruits and grapes avoid chronic inflammation by modifying the IM [80]. 

Recently, new evidence has arisen with respect to the relationship between the pathogenesis of AD and insulin resistance; it is important to consider type-2 diabetes as an essential risk factor for the formation of β-amyloid deposits in the brains of patients with dementia. In the U.S., middle-aged persons with obesity have a three-times greater risk of AD and a five-times greater risk of dementia induced by cerebral thrombosis [81]. *Moringa oleifera*, a crucifer, presents hypoglycemic effects in subjects with diabetes compared with healthy subjects [82]. In a transversal study conducted on 2031 participants, patients who consumed chocolate, wine, or tea presented significantly better scores in the cognitive state; the average ingestion of chocolate was 3.8 g/day, of wine 22 mL/day, and of tea 222 mL/day [83]. 

Hesperidin, naringin, nobiletin, and rutin (all flavonoids), resveratrol (polyphenol), and β-cryptoxanthin (carotenoid) exert anti-inflammatory and antimetabolic-syndrome effects; the latter demonstrated in humans as well as in mice. The metabolic syndrome alters the circadian rhythm because some intestinal microbes produce hormones related to sleep, such as melatonin; thus, dysbiosis induces insomnia caused by the lack of melatonin [84]. Saponins are the main components of many botanicals and traditional Chinese medicines, such as ginseng, platycodon, licorice, and alfalfa. They have poor bioavailability but can be transformed into secondary glycosides and aglycones by intestinal microbiota, further being absorbed. Based on in vivo and in vitro data, we found that saponins and their secondary metabolites have a preventive effect on metabolic syndrome (MetS). Intestinal targets involve pancreatic lipase, dietary cholesterol, and intestinal microbiota. Other targets include central appetite, nuclear receptors are transcription factors that regulate a myriad of biological processes, including cell growth and development, metabolism, reproduction, and inflammation such as PPAR (PPARs regulate gene expression by binding with retinoid X receptor as a heterodimeric partner to specific DNA sequences, termed PPAR response elements) and LXR (promotes lipogenesis, whereas PPARα controls a variety of genes in several pathways of lipid metabolism), AMPK (are enzymes that regulate the biological activity of proteins via the phosphorylation of specific amino acids with ATP as the source of phosphate, thereby inducing a conformational change from an inactive to an active form of the protein signaling pathway and adipokines levels) [85]. 

### 2.5. Cancer

Cancer is the second most frequent cause of death in industrialized countries, and nutrition is the main external factor that modulates the risk of suffering from it, contributing approximately one-third of this to the presence of all cancer types [4]. In many plant extracts, their efficacy as chemopreventives and chemotherapeutics has been assessed [6]; the phytosterols appear to play an important role in the regulation of serum cholesterol, and they present anticancer properties; similarly, the consumption of carotenoids is inversely associated with the recurrence of and death due to breast cancer [4]. Likewise, the polyphenols produce apoptosis and inhibit the proliferation of hepatocarcinoma cells [52].

Piperin is a black-pepper compound. It inhibits the glucuronidation of several chemopreventive compounds, which increases their bioavailability, rendering it a potent inhibitor of the metabolism of drugs [86]. Piperin at a dose of 75–150 μM inhibits the growth of several colon–cell cell lines; detention of the cellular cycle and of the apoptosis that occurs after the treatment with piperin provide evidence that it can be useful in the treatment of cancer of the colon [87]. The continuous support of epidemiological studies fosters the use of natural products (curcumin, capsaicin, green tea, and resveratrol) as anticancer agents; however, many of the studies present limitations; therefore, better-controlled epidemiological studies must firmly define the role of nutraceuticals in the prevention of cancer [88]. 

Breast cancer is a common cancer that occurs due to different epigenetic alterations and genetic mutations. Various epidemiological studies have demonstrated an inverse correlation between breast cancer incidence and flavonoid intake. These flavonoids can induce the expression of different tumor-suppressor genes that may contribute to decreasing breast cancer progression and metastasis. The anti-tumor effect of dietary flavonoids has been sustained by laboratory experiments, but epidemiological studies with breast cancer risk remained inconsistent and insufficient [89]. Vinblastine and vincristine are natural alkaloids with antineoplastic properties isolated from the plant *Catharantus roseus*. These alkaloids are effective anticancerous chemotherapeutic drugs that are widely used for the treatment of leukemia, lymphoma, lung carcinoma, osteosarcoma, breast cancer, gastric cancer, and ovarian cancer. Vinblastine causes the dysfunction of cellular proliferation. Oleocanthal, a phenolic compound present in extra-origin virgin olive oils, inhibits proliferation, migration, and invasion of both breast and prostate cancer cells via the inhibition of c-Met phosphorylation. Olive oil inhibits cell cycle progression and cell proliferation, induces oxidative stress and apoptosis, and stimulates the immune system, thereby preventing carcinogenesis. A combination of multiple phytochemicals and chemotherapeutic drugs can also induce a synergistic effect [90]. Berberine hydrochloride, an alkaloid, possessess anti-inflammatory, antibacterial, antidiabetic, antioxidant, sedative, and antipyretic properties. An elevated level of the expression of HIF-1 was found to be correlated with resistance to radiotherapy and unfavorable prognosis in esophageal cancer. Berberine treatment caused significant downregulation in the expression of vascular endothelial growth factor (VEGF) and HIF-1, which led to reversed radiation resistance [91]. The anticancer activities of saponins include anti-proliferation, anti-metastasis, anti-angiogenesis, and reversal of multidrug resistance (MDR). These effects are brought about by the induction of apoptosis, promotion of cell differentiation, immune-modulatory effects, bile acid binding, and amelioration of carcinogen-induced cell proliferation. Chemoprevention is the use of a chemotherapeutic agent to halt or restrict tumor development before the onset of cellular invasion. Dammarane triterpenoid isolated from *Cyclocarya paliurus* mediates anti-inflammatory activity by lowering TNF-α, PGE2, and IL-6 expression [92]. Cachexia is a multifactorial syndrome and is associated with many acute and chronic conditions resulting in compromised quality of life and increased mortality. Phytochemicals influence cancer cell behavior during proliferation, migration, and apoptosis. Flavonoids affect the NF-kB signaling pathway; administration of curcumin reduced the loss of BMI and preserved the mass of epididymal fat, gastrocnemius muscle, and anterior tibialis muscle. Green tea (*Camellia sinensis*) has been evaluated for its antioxidant and cancer chemopreventive properties. Epigallocatechin-3-gallate (EGCG) is considered the major bioactive component present in the catechin fraction of green tea; EGCG inhibits tumor cell proliferation and stimulates apoptosis, as well as prevents the angiogenesis and synthesis of cytokines, promotes muscle regeneration via activation of muscle stem cells and stimulation of their differentiation [93].

### 2.6. Infection by Helicobacter pylori

*Helicobacter pylori* is a Gram-negative bacterium. It affects more than one half of the world population, and it is an etiologic agent of the peptic acid disease in 10–20% of infected individuals, while 1–2% of these persons entertain the risk of developing gastric cancer or mucosa-associated lymphoid tissue (MALT); it exclusively colonizes the gastric mucosa [94]. Abundant sulforaphane (SFN) in crucifers induces a variety of antioxidants in the cells and organs against oxidative stress, which can contribute to the chemoprotection; in the GI tract of mice, it induces antioxidant enzymes and protects the GI mucosa against damage conditioned by *Helicobacter pylori* and anti-inflammatory non-steroidal substances (NSAIDS) [95]. Infection by *Helicobacter pylori* induces chronic oxidative stress in the gastric mucosa, which eventually favors the development of gastric cancer (Figure 5); in a study, mice and humans were given broccoli and alfalfa during 8 weeks, observing a significant reduction in pepsinogen I and II (marker of atrophic gastritis that predisposes to the development of gastric cancer) [*p* < 0.05] compared with the baseline value. It returned to baseline values 2 months after the intervention. During this study, the oral ingestion of SFN diminishes the inflammation of the gastric mucosa due to the infection by *Helicobacter pylori* as well as to the consumption of NSAIDS [96] by means of the cytoprotective and antioxidant effect and through the activation of the anti-inflammatory cells via the Nrf-2 transcription factor [95,97]. The majority of the studies report the in vitro efficacy against *Helicobacter pylori* of the therapy with herbs, but this is not always followed by an effective eradication of the bacteria in animal models and/or in clinical studies; the incapacity of the compounds can condition the latter to resist the acid medium of the stomach, and to their incapacity to reach the bacteria through the mucosal layer, due to the use of an insufficient dose [98]. The flavonoids, as well as resveratrol, inhibit the growth of the bacteria by inhibiting the urease enzyme; curcumin suppresses the expression of the matrix metalloproteinases-3 and -9, inflammatory molecules associated with the pathogenesis of *Helicobacter-pylori* infection [99,100]. 

Glabridin, an isoflavone, has a wide variety of biological effects (antioxidant, anti-inflammatory, antioxidant, anti-atherogenic, regulation of energy metabolism, estrogenic, neuroprotective, and anti-osteoporotic) [101]. The oregano and cranberry extract mixture (25% oregano and 75% cranberry) was superior in inhibiting *H. Pylori*; this may be due to one, or more than one, phenolic compound damaging the cell membrane. Oregano and cranberry are useful botanicals that are known for their antimicrobial activity linked to the phenolic moiety; both cause urease inhibition. Phenolic phytochemicals such as ellagic acid and rosmarinic acid have the potential to interact with proteins and alter their conformation. These phytochemicals can directly interact with the receptors on the cell membrane and could affect the normal functioning of ion pumps; as a consequence, impairment of proton pumps and loss of H^+^-ATPase in damaged membranes can cause disruption in the normal cellular function of the microorganism and, therefore, lead to cell death [102]. The gastroprotective and anti-H. pylori effects of methanolic extract (ME) from A. Triplinervia; the results showed that the minimum inhibitory concentrations (MIC) of ME against *H. pylori* were 0.25 mg/mL, ME presents excellent antimicrobial action against one of the most important factors that cause gastric ulceration. Active constituents responsible for the gastroprotective action are concentrated in the ethyl acetate fraction (EAF) (50% protection) rather than in the aqueous fraction, which did not induce significant gastroprotection at the same dose (100 mg/kg). EAF induced an increase in gastric mucosa PGE2 levels, which remained high even after the previous administration of indomethacin. The phytochemical profile of ME revealed that EAF contains mainly flavonoids [103]. The effect of *Camellia sinensis* extracts, polyphenolic compounds, and catechin contents (epicatechin, epigallocatechin, and epicatechin gallate) on the urease enzyme is a major colonization factor for *H. Pylori.* A concentration of 4 mg/mL of nonfermented and 5.5 mg/mL of semifermented extract are bactericidal for *H. Pylori*; *Camellia sinensis* extracts, especially the nonfermented, could reduce *H. pylori* population and inhibit urease production at lower concentrations [104]. 

The crude methanol extract of the leaf of allium known as garlic, phytochemical screening of the plant showed that it contains alkaloids, cardiac glycosides, and saponins that decrease urease activity [105].

### 2.7. Diabetes 

Diabetes mellitus (DM) is a public health problem [106]. It affects the vital organs; nearly 40% of patients acquire diabetic nephropathy (DN), which leads to a renal system dysfunction that, if not reverted, leads to a terminal kidney disease. The factors that affect DN mainly include elevated ROS levels, persistently elevated glucose levels, the increase in proinflammatory cytokines, the advanced glycation end products (AGE), and alterations at the molecular level, such as altered protein kinase B (PKB), activated in turn by the adenosine-monophosphate-activated kinase (MAPK). Metabolic memory is a synonym of glycemic memory, an innate mechanism in which diabetic complications continue to worsen even after achieving good glycemic control. Hyperglycemia negatively affects all kidney cells, producing AGE and increasing the levels of growth factors such as angiotensin II and the transforming growth factor-beta 1 (TGF-(1)) in the kidney cells. The TGF plays an important role in DN because it possesses profibrotic actions [107]; some medicinal plant phytochemicals help reduce the risk of DN, for example, silybin, deriving from *Silybin marianum* [108], *Curcuma longa*, *Berberis vulgaris*, and *Andrographis paniculata*, in which the antioxidants present in these provide protection to the nephrons on diminishing the oxidative stress [109]; the flavonoids not only act on the inflammation pathway but also contribute to low blood glucose levels, thus reducing the risk of diabetes-related diseases [106]. *Abroma august* is widely utilized in the treatment of diabetes and its complications in reducing vascular inflammation, hyperglycemia, and oxidative stress, with its extract negatively regulating the expression of IL-1β, NF-κB, IL-6, and TNF-α, suggesting its capacity as a nephroprotective agent [110]. Rutin significantly diminishes the levels of glucose, creatinine, ureic nitrogen, and proteins in the urine and can also improve oxidative stress, inhibit the accumulation of type-IV collagen, and reduce the levels of AGE and TGF-β1. Rutin can be an effective drug for the prevention of early DN. Rutin is one of the flavonoid’s greatest utilities against diabetes; it has been shown to reduce fasting blood glucose, improve glucose tolerance, and also reduce serum lipids more effectively [15,106,111]. *Bebincad ceridera* prevents lipid peroxidation caused by oxidative stress; catechin prevents the development of diabetes and complications associated with it, and similarly, curcumin has multiple nephroprotective effects due to its antioxidant property; soy delays the progression of DN; *Vitis vinifera* has resveratrol, which reduces kidney dysfunction; betanin obtained from beets, reduces glomerulosclerosis, the glomerular surface area, and tubulointerstitial fibrosis [112].

In a meta-analysis that included six prospective studies, 284,806 participants were identified, including 18,146 cases. The relative risk (RR) of type-2 diabetes for those with a high ingestion of flavonoids compared with a low ingestion was 0.91 (95% CI 0.87–0.96); in addition, an increase in the consumption of 500 mg/day was associated with a significant risk reduction of 5% (RR = 0.95, 95% CI 0.91–0.98) [113]. Likewise, in 2015, the possible beneficial effect was evaluated of the consumption of β-carotenes in the development of type-2 diabetes; in this study, 37,846 participants were included, with an average ingestion of 10 ± 4 mg/day and an average follow-up of 10 ± 2 year, with 915 incident cases reported. After carrying out the adjustment for age, gender, risk factors for diabetes, dietary intake, waist circumference, and BMI, higher ingestion of β-carotenes was inversely associated with the risk of diabetes [114]. 

Quercetin reduces GLUT4 activity and diminishes gluconeogenesis and hepatic glycogenolysis; combining quercetin with sitagliptin improves β-cell function and glycemic control, as well as the metabolic profile [15,106]. Piperin promotes glucose uptake in the skeletal muscles; treatment of the rates of diabetes with piperin plus quercetin gives rise to a significant reduction in the concentration of blood glucose [115]. Lipid peroxidation and oxidative stress are the causes of the development of DM; in a similar fashion, dysbiosis increases the risk of DM in that a diet rich in fats increases up to three times the production of LPS, thus favoring the inflammation process and insulin resistance [116]. Intestinal dysbiosis also affects the production of SCFA, an important point in terms of the integrity of the intestinal barrier, the proliferation of the pancreatic cells, and the biosynthesis of insulin [117]. 

Adipokines, which are biologically active molecules secreted from adipose tissues or adipocytes, play major roles in the regulation of food intake, insulin sensitivity, energy metabolism, and the vascular microenvironment; they are involved in the obesity-induced chronic inflammatory response that plays a crucial role in the development of obesity-related pathologies such as type II diabetes and atherosclerosis. The dysregulation of adipokine release, which causes obesity-induced inflammation, is the common denominator that links obesity to the pathogenesis of insulin resistance and atherosclerosis. Capsaicin (an alkaloid) decreased the levels of IL-6 and MCP-1 and increased the level of adiponectin released from obese fat tissues and fat cells. This dual action of capsaicin may be favorable for improving the dysregulation of adipokine release through the normalization of the balance between the secretions of proinflammatory and anti-inflammatory adipokines in insulin resistance or atherosclerosis [118]. Ginseng is one of the most valuable and commonly used Chinese medicines, not only in ancient China but also worldwide. Ginsenosides, also known as saponins or triterpenoids, are thought to be responsible for the beneficial effects of ginseng. The anti-diabetic effect of ginseng is positive for type-2 diabetic patients but has no significant impact on prediabetes or healthy adults [119]. Ginsenoside Rb1 exerts various pharmacological effects on metabolic disorders, including the attenuation of glycemia, hypertension, and hyperlipidemia, which depend on the modulation of oxidative stress, inflammatory response, autophagy, and anti-apoptosis effects by regulating the effects of glycolipid metabolism and improving insulin and leptin sensitivities [120].

### 2.8. Bone Diseases 

Osteopenia is a metabolic disorder that consists of a reduction in bone density. If undetected and mistreated, the gradual loss of calcification can eventually result in osteoporosis (OP) and fractures. It is a significant and impactful public health issue, affecting more than 20 million elderlies in the US, especially postmenopausal (PM) women, and it causes about 1.5 million fractures annually [121]. In the developed world, 2–8% of males and 9–38% of females are afflicted with osteoporosis [4]; however, this prevalence has increased according to the diagnostic criteria of the World Health Organization (W.H.O.); the prevalence of osteoporosis is greater in patients in developing countries (22.1%; 95% CI 20.1–24.1%) than in developed countries (14.5%; 95% CI 11.5–17.7%), and its prevalence varies substantially between countries and regions [122]. Osteoporosis becomes more prevalent common with age and is more prevalent in females than in males [4]. In postmenopausal women, the hormonal changes linked to ovarian function after menopause cause the majority of the alterations in bone density. The estrogen receptor (ER) and androgen receptor (AR) that are expressed in osteoblast and osteoclast progenitors and their offspring influence the role of estrogens and androgens on bone mass. Estrogens increase the longevity of osteoblasts (OB) and osteocytes while decreasing that of osteoclasts, limiting skeletal turnover [123]. Oxidative stress accelerates bone remodeling and bone mass loss, whereas antioxidants reduce oxidative stress and stop bone mass loss. The loss of estrogens or androgens, similar to old age, increases reactive oxygen species (ROS) in bone cells. In contrast, systemic administration of antioxidants attenuates bone loss due to sex steroid deficiency in male and female mice [124]. 

Soy isoflavones significantly increase bone mineral density by 54%; this was evidenced in a meta-analysis of 19 studies in which the association was investigated of soy isoflavones and the risk of osteoporosis (odds ratio [OR] = 0.54; 95% CI 0.13–0.94) [125] in a similar way, they can prevent osteoporosis in postmenopausal women and the risk of fractures [4]. It is known that nutrition and physical activity play an important role in skeletal health, achieving the highest bone mineral density (BMD) and maintaining bone health. In this review, the role of macronutrients (proteins, lipids, and carbohydrates), micronutrients (mineral–calcium, phosphorus, magnesium, as well as vitamins D, C, and K), and flavonoid polyphenols (quercetin, rutin, luteolin, kaempferol, and naringin), which appear to be essential for the prevention and treatment of osteoporosis; these last are also involved in bone formation [126,127]. Dietary quercetin (2.5% for 4 weeks) can increase BMD and improve cortical and trabecular bone microstructure in mice [128]. Following quercetin administration, enhanced BMD, trabecular bone microarchitecture, and improved bone strength have also been determined in OVX rats (50 mg/kg/day for 8 weeks) [129]. Ginsenoside compound K (GCK) could influence the occurrence and progress of osteoporosis by interacting with 138 potential target proteins, of which 16 targets act in the osteoclast differentiation pathway [130]. β-carotene may improve BMD and reduce the risk of osteoporosis and fracture (95% Cl (−0.391, −0.034), I^2^ = 87.30%, *p* = 0.019]); subgroup analysis showed that the intake of β-carotene was negatively associated with the risk of osteoporosis in both the male subgroup [OR = 0.7, 95% Cl (0.549, 0.893), I^2^ = 40.40%, *p* = 0.004] and female subgroup [OR = 0.684, 95% Cl (0.487, 0.960), I^2^ = 86.40%, *p* = 0.028] [128]. Anti-osteoporosis alkaloids are classified into six categories: they promote mesenchymal stem cell differentiation, improve osteoblast proliferation, stimulate osteoblast autophagy, suppress osteoclast formation, regulate multiple signaling pathways, including interrupting the tumor necrosis factor receptor-associated factor 6- receptor activator of NF-κB interaction, inhibiting the NF-κB pathway in osteoclasts, activating the p38 mitogen-activated protein kinases pathway in osteoblasts, and triggering the wingless and int-1 pathway in mesenchymal stem cells. The natural alkaloids exert anti-osteoporotic effects mainly by inhibiting bone resorption [129]. Osteoarthritis (OA) is the leading cause of disability for millions of people; there is no effective treatment for OA, only the therapy that slows or prevents OA progression. Studies show that rosehip berries, ginger root, green tea leaves, turmeric root, and pomegranate peel, containing a significant amount of polyphenols and phytoflavonoids have chondroprotective activity, which positively affects OA prevention and treatment [131]. Daily consumption of green tea extracts slowed the progression of arthritis in rats, suppressed serum IL-17 levels, and increased serum IL-10 levels [132,133]. 

### 2.9. Kidney Disease 

It has been reported that phytochemicals (catechin, epicatechin, epigallocatechin-3-gallate, diosmin, rutin, quercetin, hyperoside, and curcumin) are useful as antioxidants and are efficient for the prevention of urolithiasis (the process of the formation of calcium stones in the urinary tract). Kidney stones exert an effect on the health system; they have a prevalence of >10% and a relapse rate of 50% in 5–10 years [134]; it has been reported that 10–12% of persons in industrialized countries (10% of males and 3% of women) present a kidney stone during their life; the etiology of this disease is multifactorial and is related with genetics, diet, and sedentarism; the kidney stones that contain calcium are the most frequent (75–90%), including hypercalciuria, hypocitraturia, hyperuricosuria, and hyperoxaluria, which are found frequently in the formers of calcium kidney stones, the latter observed as being influenced therapeutically, this formation appearing to be the result of excessive consumption of protein. These four factors favor the process of crystallization in urine; however, the urine is habitually protected in the nucleation, growth, and aggregation of calcium minerals through crystallization inhibitors. In the urine, calcium oxalate crystallization can only be induced by extreme oversaturation and deficient activity of crystallization promoters [135]. Dietary fiber, which is abundant in fruits and vegetables, can diminish the formation of kidney stones due to the non-digestible ingredients that unite with the fats and minerals of the GI digestive tract, resulting in the urinary excretion of oxalate and calcium. Calcium in the diet can act on the uniting of the dietary oxalate in the intestine, which reduces the absorption of oxalate and the urinary excretion of it [136,137]. 

Green tea (*Camellia sinensis*) contains many polyphenols, which are protector-effect antioxidants against the development of kidney stones; likewise, its supplementation inhibits the growth of crystals in rats, diminishing the growth of crystals in the rat kidney, diminishing the excretion of oxalate [138,139]. Raspberries possess polyphenols and inhibit the formation of stones due to the presence of citrate, magnesium, and glycosaminoglycans [140]. *Rubia cordifolia* is effective in the treatment of different kidney diseases and exerts preventive effects on the formation of urinary stones, inhibits the excretion of calcium, prevents hyperoxaluria and hypocitraturia on diminishing the formation of urinary oxalate, and regulates the reabsorption of tubular citrate, respectively. The nephroprotective effect of these plants could be attributed to their antioxidant properties [141]. The persimmon (*Punica granatum*), employed in traditional medicine, is a rich source of polyphenols, alkaloids, and anthocyanins, which are highly capable of eliminating free radicals. Phytochemicals produce relaxation of the urinary tract; consequently, the stone can be easily eliminated from the kidney [142]. 

Polycystic kidney disease (PKD) is inherited and is characterized by kidney cysts; it is one of the principal causes of terminal kidney disease. The polyunsaturated fatty acids can modify multiple steps in the pathogenesis of PKD. Phytoestrogens and phytochemicals affect the development of physiological events. Rats fed with soy proteins with a high content of isoflavones favor a greater reduction in the inflammation process, conserve normal kidney function, and reduce changes in the cysts, compared with soy products with a low content of isoflavones [143]. It has been informed that curcumin exhibits beneficial effects against inflammatory diseases; cells treated with curcumin reduce the epithelial proliferation of cysts and prevent the development of renal insufficiency [144,145]. Parsley (*Petroselinum crispum*), belonging to the family *Umbelliferae*, is commonly known as an herb, spice, and vegetable and is widely distributed in Western Asia, the Mediterranean, and several European countries. These beneficial activities could be due to its bioactive constituents, including flavonoids, carotenoids, coumarins, tocopherol, and ascorbic acid. Parsley and its extracts have been used potentially as a complementary/alternative treatment for various renal diseases. *P. crispum* has been used as a promising anti-urolithiasis remedy. Its ethanolic extract prevented the nucleation and precipitation of calcium oxalate, urine supersaturation, and urinary protein excretion in a rat model of calcium stone formation [137]. 

### 2.10. Cardiovascular Disease (CVD) 

It is estimated that 90% of CVD can be prevented with a healthy diet, physical exercise, avoiding the consumption of tobacco, and limiting the consumption of alcohol. Using a systematic review, the association was evaluated of the daily ingestion of flavonoids and the risk of CVD, determining that the increase of every 10 mg/day of flavonoids is associated with a 5% diminution in the development of CVD [146]; Zhou et al. demonstrated the association between the consumption of flavonoids and the reduction in the risk of mortality (hazard ratio [HR]: 0.87; 95% CI 0.81–0.94) in women aged 50 years and over [147]. An increase in the consumption of flavonoids of 20 mg/day was associated with a risk reduction of 14% for the development of a cerebrovascular accident [4]. Chrysin is a flavone; it reduces right-ventricle systolic pressure and the mean pressure of the pulmonary artery, diminishing in the same manner the expression of collagen type-I and -III and of NF-κB, thus preventing pulmonary hypertension in the rat model [148]. Morbidity and mortality from acute myocardial infarction (AMI) remain substantial, although interventional coronary reperfusion strategies are widely used and successful. MI remains the most common cause of heart failure (HF) worldwide. Panax notoginseng saponins (PNS), the extract of Panax notoginseng, exerts a cardioprotective effect in AMI, and the underlying mechanism refers to inducing cardiomyocyte autophagy, antiplatelet aggregation, enhancing endothelial migration and angiogenesis [149]. Berberine is an alkaloid found in such plants as Berberis; in vitro, it exerts significant anti-inflammatory and antioxidant activities. In animal models, berberine has neuroprotective and cardiovascular protective effects. In humans, its lipid-lowering and insulin-resistance-improving actions have clearly been demonstrated in numerous randomized clinical trials. Moreover, preliminary clinical evidence suggests the ability of berberine to reduce endothelial inflammation, improving vascular health, even in patients already affected by cardiovascular diseases [150]. Epidemiological studies have demonstrated that polyphenol-rich foods reduce cardiovascular events in the general population and patients at risk of cardiovascular diseases [151]. Kaempferol, quercetin, and resveratrol prevent oxidative stress by regulating proteins that induce oxidation in heart tissues. In addition, polyphenols modulate the tone of the endothelium of vessels by releasing NO and reducing LDL oxidation to prevent atherosclerosis. In cardiomyocytes, polyphenols suppress the expression of inflammatory markers and inhibit the production of inflammation markers [152]. 

## 3. Conclusions and Future Perspectives 

There are multiple pathways implicated in the pathogenesis of diseases, one of these being oxidative stress induced by a disequilibrium between the production of reactive oxygen species and reactive nitrogen species; phytochemicals have attracted increasingly more attention as potential agents of the regulation of oxidative stress, as well as of the release of free radicals, inflammation, the immune response, insulin resistance, lipid metabolism, and the intestinal microbiota, the scientific bases that have opened the possibility of these being utilized in the prevention and treatment of various diseases. The intestinal microbiota has important functions for health since different metabolites with beneficial potential are produced by digesting dietary products. There is solid evidence that supports the role of the microbiome in the development and progression of diseases, and this microbiome is modified (dysbiosis) by the diet; therefore, the consumption of phytochemical products aids in the maintenance and restoration of health. Future investigations are necessary to evaluate the bioavailability, metabolism, and efficiency of phytochemicals in health; because of this, in spite of knowing their metabolism and the amount of some phytochemicals that remain in the organisms, little is known regarding the doses that humans can consume, as well as the potential adverse effects that these can condition at these doses. 

## Figures and Tables

**Figure 1 plants-13-00523-f001:**
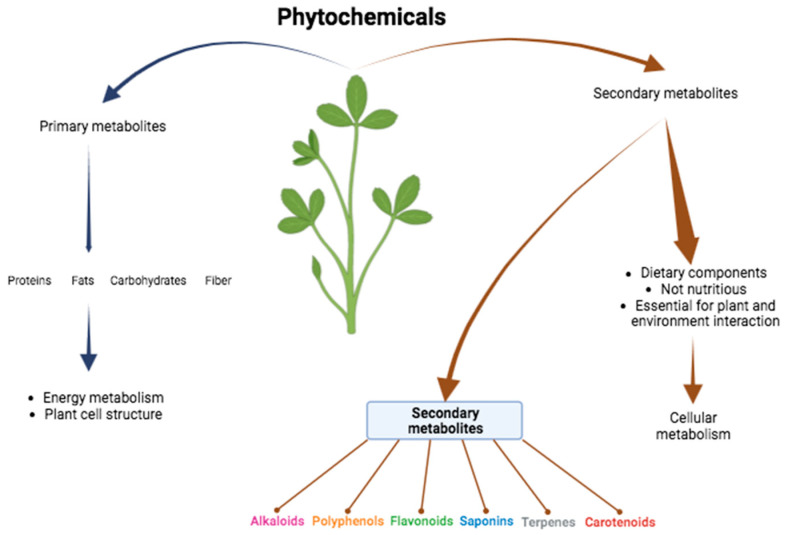
The difference between primary and secondary metabolites. Created with BioRender.com.

**Figure 2 plants-13-00523-f002:**
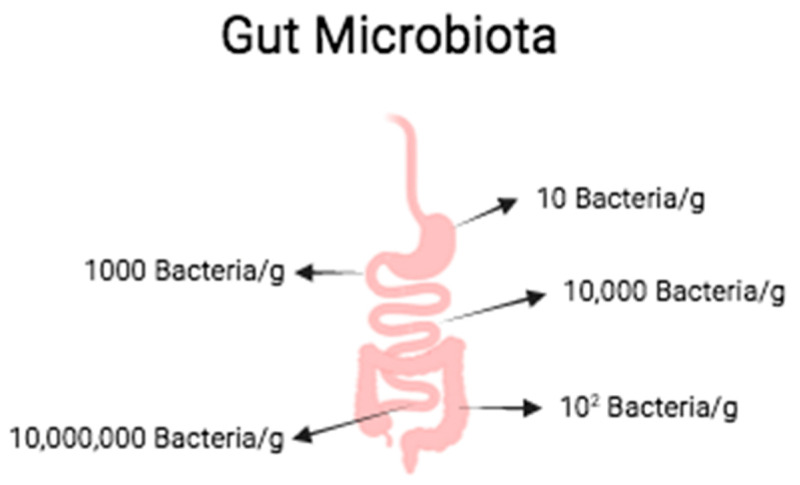
Normal intestinal microbiota. Created with BioRender.com.

**Figure 3 plants-13-00523-f003:**
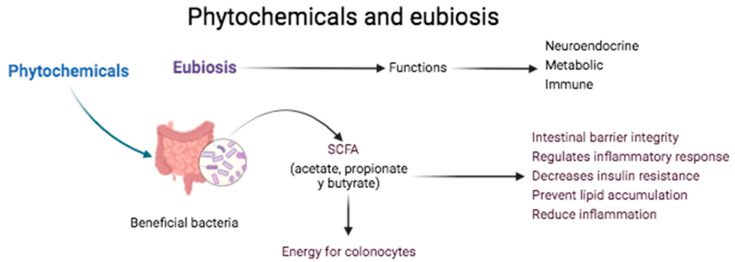
Effects of phytochemicals on the intestinal microbiota. Created with BioRender.com.

**Figure 4 plants-13-00523-f004:**
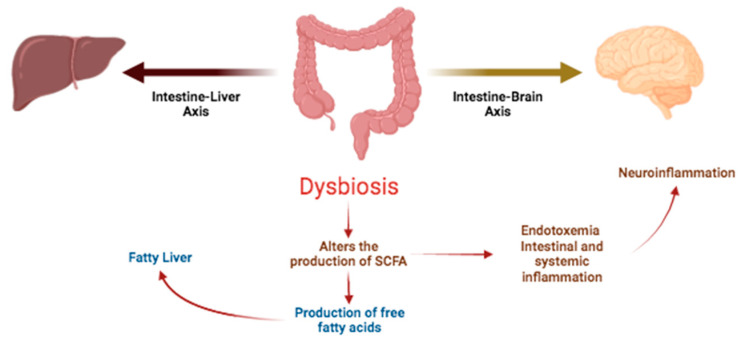
Role of the intestinal microbiota in intestine–liver and intestine–brain axis. Created with BioRender.com.

**Figure 5 plants-13-00523-f005:**
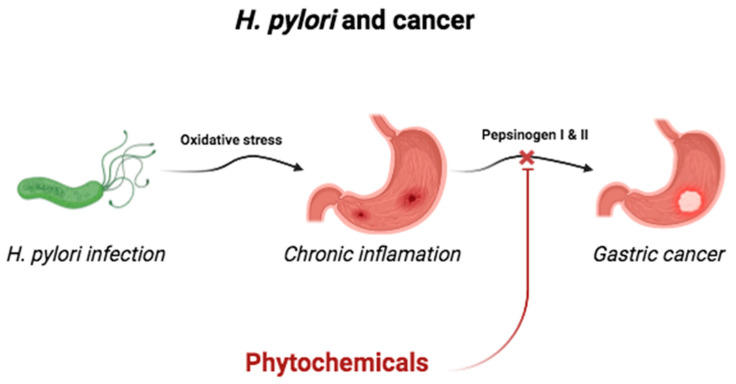
Complication associated with *Helicobacter pylori* infection. Created with BioRender.com.

## Data Availability

Data are contained within the article.

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
