# Peer review of "Phytochemicals and Their Usefulness in the Maintenance of Health"

_plants, 2024, doi:10.3390/plants13040523_

Round 1
Reviewer 1 Report
Comments and Suggestions for Authors
The proposed article explores numerous phytochemicals and their significance in maintaining health, with a strong emphasis on physiology and the interaction between phytochemicals and the body's organs and processes. On the basis of a careful reading of the manuscript, I decided that the manuscript in its present form is suitable for publication after corrections.
The main limitations of the manuscript are as follows:
- Essential oils, which are crucial and active phytochemicals, were not cited in the paper.
- The text requires both language and format revisions, as numerous instances of double spaces were found throughout the document.
- The abstract needs to be completely rewritten as it does not reflect the content of the paper.
- Line 57-62: The sentences has grammatical issues
- Lines 73-74: the text is a little confusing: “the phytochemical groups include carotenoids, common in vegetables and fruits, containing 50 of the 700 natural carotenoids important.”
- Lines 72-89: All the data cited in this part are from unknowns sources. Please add references for these statements
- Line 104: Please complete the reference correctly “Martel et al.,”
- Line 110: Remove “for”
- Lines 128-134 : this paragraph should be removed
- Line 250: “In terms of Akkermansia muciniphila, its effect has been studied on intestinal health.” Please add reference.
- Lines 634-642: The passage has some grammatical issues and should be revised for clarity
- Lines 744-745: The sentence has grammatical issues “The intestinal microbiota affects health, in that on the digesting the dietary products.”

Comments on the Quality of English Languageacceptable
Author Response
Reviewer 1
Information was added in the different sections and the reference was marked yellow
The main limitations of the manuscript are as follows:
Essential oils, which are crucial and active phytochemicals, were not cited in the paper.
- Answer: Added the following references
- Svoboda KP.; Deans SG. Biological Activities of Essential Oils from Selected Aromatic Plants. Acta Horticulturae, 1995, 390: 203-209.
- Sharifi-Rad J.; Sureda A. Biological Activities of Essential Oils: From Plant Chemoecology to Traditional Healing Systems. 2017, 11, 22: 1-55.doi: 10.3390/molecules22010070
- Ramsey JT.; Shropshire BC. Essential Oils and Health. Yale J Biol Med. 2020 , 93(2): 291-30
- The text requires both language and format revisions, as numerous instances of double spaces were found throughout the document: Answer: Checked grammar and removed double spaces - The abstract needs to be completely rewritten, as it does not reflect the content of the paper: Answer:was rewritten- Line 57-62: The sentences have grammatical issues: Answer:grammar was corrected - Lines 73-74: the text is a little confusing: “the phytochemical groups include carotenoids, common in vegetables and fruits, containing 50 of the 700 natural carotenoids important.” Answer:the text was corrected: The classification of phytochemicals is based on their chemical structure and on their functional characteristics, and the phytochemical groups include carotenoids, common in vegetables and fruits, containing 50 of the 700 natural carotenoids important in human nutrition - Lines 72-89: All the data cited in this part are from unknown sources. Please add references for these statements: Answer:references were included: references number: 4,10-12 - Line 104: Please complete the reference correctly “Martel et al.,” Answer:The text was modified and the bibliography was left with its corresponding reference number. - Line 110: Remove “for”:Answer: removed
- Lines 128-134: this paragraph should be removed:
Answer: deleted
- Line 250: “In terms of Akkermansia muciniphila, its effect has been studied on intestinal health.” Please add reference:Answer: reference number 35 was added
Derrien M.; Belzer C. Akkermansia muciniphila and its role in regulating host functions. Microb Pathog. 2017, 106: 171-181. doi: 10.1016/j.micpath.2016.02.005
- Lines 634-642: The passage has some grammatical issues and should be revised for clarity:Answer: Revised and corrected grammar: Rutin can be an effective drug for the prevention of early DN. Rutin is one of the flavonoids greatest utility against diabetes, it is shown to reduce fasting blood glucose, improve glucose tolerance, and also reduce serum lipids more effectively. Bebincad ceridera prevents lipid peroxidation caused by oxidative stress; Catechin prevents the development of diabetes and complications associated with it, and similarly, curcumin has multiple nephroprotective effects due to its antioxidant property; soy delays the progression of DN; Vitis vinifera has resveratrol which reduces kidney dysfunction; Betanin obtained from beets, reduces glomerulosclerosis, the glomerular surface area, and tubulointerstitial fibrosis - Lines 744-745: The sentence has grammatical issues “The intestinal microbiota affects health, in that on the digesting the dietary products.” Answer: was reviewed and corrected (The intestinal microbiota has important functions for health, since by digesting dietary products; different metabolites with beneficial potential are produced.)
Reviewer 2 Report
Comments and Suggestions for Authors
Plants are a rich source of secondary metabolites, 43 commonly denominated phytochemicals that are grouped as alkaloids, polyphenols, flavonoids, saponins, carotenoids, and terpenes, with diverse structural and functional properties. In this paper, authors reviewed the application of phytochemicals in the treatment of intestinal microbiota, inflammatory bowel disease, liver, metabolic syndrome, cancer, infection by helicobacter pylori, osteoporosis, kidney disease, cardiovascular disease and the functional mechanism.
1. Firstly, the abstract did not reflect the content of the review. The abstract must be revised more accurate.
2. The major phytochemicals components (such as alkaloids, polyphenols, flavonoids, saponins, carotenoids, or terpenes) should be mentioned when they were used in different disease.
3. The “cancer” in the Function of the phytochemicals in the organism part should be more detail because it is attracted topical.
4. There are some writing and formatting problems, such as in line 36, subscript 2 for O2· and H2O2; line 162, it is multiple sign “×”, not letter “x”.
Author Response
Reviewer 2. Information was added in the different sections and the reference was marked green
- Firstly, the abstract did not reflect the content of the review. The abstract must be revised more accurate:
Answer:was rewritten
2. The major phytochemicals components (such as alkaloids, polyphenols, flavonoids, saponins, carotenoids, or terpenes) should be mentioned when they were used in different disease: Answer:The different phytochemicals were added in each of the different diseases, the missing phytochemicals were added in each section. (references: 43,64,78, 82-84,86, 94-98, 111-113, 116-121, 138-141) 3. The “cancer” in the Function of the phytochemicals in the organism part should be more detail because it is attracted topical:Answer: information was added (references: 89-93)
- There are some writing and formatting problems, such as:
- in line 36, subscript 2 for O2· and H2O2: Answer:•O2, H2O2
- line 162, it is multiple sign “×”, not letter “x”: nearly 4 ´ 1013,
Reviewer 3 Report
Comments and Suggestions for Authors
In this review, the authors address the importance of active ingredients of plant origin in the maintenance of health and prevention of disease.
The following comments on the work:
Part Introduction needs to be better organized - the description is very cursory; it does not introduce the reader to the topic of the publication and does not clearly and legibly explain the purpose of the work.
The first part (Lines 72-103) regards the phytochemicals, which are the main subject of this article, and is described very poorly and briefly.
Furthermore, the Abstract indicates the importance of phytochemicals in the reduction of oxidative stress, whereas the Introduction, after a cursory characterization of various phytochemicals, only discusses the importance of polyphenolic compounds as antioxidants - also very briefly.
Moreover, to further discuss the usefulness of phytochemicals in maintaining health, it would be good to indicate firstly the mechanisms of antioxidant action of phytochemicals.
In the following section of the paper, in which the authors address the important topic - the usefulness of phytochemicals in maintaining, it can be concluded that the description and examples of phytochemicals and scientific studies cited are limited; this is not a complete review of the literature, only selected examples are cited, in some subsections e.g. Osteoporosis only soy isoflavones, while for example, PubMed base shows about 200 results.
Importantly, for the paper to be of greater scientific value, scientific studies relating to the importance/use/effectiveness of the phytochemicals mentioned should be cited in the individual chapters of the paper: Intestinal Microbiota; Inflammatory Bowel Disease; Liver; Metabolic Syndrome; Cancer; Infection by Helicobacter pylori; Osteoporosis; Kidney Disease; Cardiovascular Disease; e.g. Line 454: Please cite the original research results from which the data on Nuciferine or Line 513: silymarin, among many others.
The review article must not be based on review papers. The results of the original research must be referenced and cited - please consider this comment about the work as a whole.
The paper would be more readable if it included a summary of the individual phytochemicals in several tables about each disease entity for each subsection - this is only suggestion from the reviewer.
In addition, please note the publications cited; I suggest citing the most recent research results where possible, from the last 5 years.
In addition to the above comments on the article as a whole, here are some specific tips/notes to be made:
1. Line 55 – [5-7] instead [5,6,7]
2. If an abbreviation has already been used and developed for the first time in the text, it should continue to be used consistently in the work without being developed again e.g. line 125 (MCP-1) and then line 151, or line 144 nitric acid synthase (iNOS) and line 344 inducible nitric acid synthase (iNO) – please check this throughout the text
3. Please use a consistent and standardized notation throughout the whole paper - nitric oxide (NO) or nitric oxide (•NO); Prostaglandin E2 (PGE2) or PGE2; Monocyte chemoattractant protein or monocyte chemoattractant protein 1; Intestinal Microbiota (IM) or Intestinal microbiota (IM) and others in the whole text
4. Line 92 „They are linked with sugars…” unclear
5. Line 93-94: „vegetables (onions, lettuce, tomatoes) and fruits (apples, grapes, citrus fruits, cranberries, strawberries, and raspberries)” – only in these vegetables and fruits?
6. Line 123: „…antiviral, and anti-…. ?
7. Line 140: „arachnid acid” ? or arachidonic?
8. Line 140 „nitric oxide species (NOS)” or maybe reactive nitrogen species (RNS)
9. Line 142: „dyosmin” or rather „diosmin”
10. Line 171: „The IM depends on the host, provides benefits for same [16], and…. – unclear
11. Line 367: Puerarin – please give examples of the plant occurrence of this compound
12. Line 356: „Liver”, maybe better „Liver disorders/ diseases” as in the other subsections, diseases are described, not individual organs
13. Line 448: in which aspect Vitamin E is described, since the paper concerns phytochemicals. The same Line 519: vit C and Line 555: vit D
14. Line 739: what the authors have in mind: “phytochemicals, and principally their bioactive compounds”? after all, phytochemicals are plant-based bioactive compounds produced by plants

Author Response
Reviewer 3 Information was added in the different sections and the reference was marked purple
Part Introduction needs to be better organized - the description is very cursory; it does not introduce the reader to the topic of the publication and does not clearly and legibly explain the purpose of the work:
Answer: was rewritten
The first part (Lines 72-103) regards the phytochemicals, which are the main subject of this article, and is described very poorly and briefly. Furthermore, the Abstract indicates the importance of phytochemicals in the reduction of oxidative stress, whereas the Introduction, after a cursory characterization of various phytochemicals, only discusses the importance of polyphenolic compounds as antioxidants - also very briefly. Answer:Information was added and the references are from number 20 to 23. . In the following section of the paper, in which the authors address the important topic - the usefulness of phytochemicals in maintaining, it can be concluded that the description and examples of phytochemicals and scientific studies cited are limited; this is not a complete review of the literature, only selected examples are cited, in some subsections e.g. Osteoporosis only soy isoflavones, while for example, PubMed base shows about 200 results:Answer: It was about bone diseases and the added references are:
- NIH Consensus Development Panel on Osteoporosis Prevention, Diagnosis, and Therapy. Osteoporosis prevention, diagnosis, and therapy. JAMA. 2001 Feb 14;285(6):785-95.doi: 10.1001/jama.285.6.785 )
- Rossouw JE.; Anderson GL. Risks and benefits of estrogen plus progestin in healthy postmenopausal women: principal results From the Women's Health Initiative randomized controlled trial. JAMA. 2002, 17, 288(3): 321-33.doi: 10.1001/jama.288.3.321
124. Almeida M.; Laurent MR. Estrogens and Androgens in Skeletal Physiology and Pathophysiology. Physiol Rev. 2017, 97(1): 135-187.doi: 10.1152/physrev.00033.2015 )
- Sukhikh S.; Noskova S. Chondroprotection and Molecular Mechanism of Action of Phytonutraceuticals on Osteoarthritis. Molecules 2021, 26: 1-18.doi: 10.3390/molecules26082391
132.Marotte H.; Ruth JH. Green tea extract inhibits chemokine production, but up-regulates chemokine receptor expression, in rheumatoid arthritis synovial fibroblasts and rat adjuvant-induced arthritis. Rheumatology (Oxford). 2010, 49(3): 467-479. doi: 10.1093/rheumatology/kep397
- Importantly, for the paper to be of greater scientific value, scientific studies relating to the importance/use/effectiveness of the phytochemicals mentioned should be cited in the individual chapters of the paper: Intestinal Microbiota; Inflammatory Bowel Disease; Liver; Metabolic Syndrome; Cancer; Infection by Helicobacter pylori; Osteoporosis; Kidney Disease; Cardiovascular Disease; e.g. Line 454: Please cite the original research results from which the data on Nuciferine or Line 513: silymarin, among many others: Answer:References marked with numbers are added: 65-67, 76,77. -The review article must not be based on review papers. The results of the original research must be referenced and cited - please consider this comment about the work as a whole. Answer:The original texts were searched, the text was modified and those references were added.
In addition to the above comments on the article as a whole, here are some specific tips/notes to be made:
1. Line 55 – [5-7] instead [5,6,7]: Answer:the change was made
- If an abbreviation has already been used and developed for the first time in the text, it should continue to be used consistently in the work without being developed again e.g. line 125 (MCP-1) and then line 151, or line 144 nitric acid synthase (iNOS) and line 344 inducible nitric acid synthase (iNO) – please check this throughout the text:
Answer:the text was revised, and only abbreviations were left, without being developed again
3. Please use a consistent and standardized notation throughout the whole paper - nitric oxide (NO) or nitric oxide (•NO); Prostaglandin E2 (PGE2) or PGE2; Monocyte chemoattractant protein or monocyte chemoattractant protein 1; Intestinal Microbiota (IM) or Intestinal microbiota (IM) and others in the whole text:Answer: the text was corrected in its entirety using a coherent and standardized notation 4. Line 92 „They are linked with sugars…” unclear: Answer:the word sugar was removed, the cited text was revised and the paragraph was modified to make it clear 5. Line 93-94: „vegetables (onions, lettuce, tomatoes) and fruits (apples, grapes, citrus fruits, cranberries, strawberries, and raspberries)” – only in these vegetables and fruits?: Answer:texts were searched and missing fruits and vegetables were added 6. Line 123: „…antiviral, and anti-…. ? Answer:the last word anti-, was an additional error, was removed from the text 7. Line 140: „arachnid acid” ? or arachidonic?: Answer:arachidonic acid, modified 8. Line 140 „nitric oxide species (NOS)” or maybe reactive nitrogen species (RNS): Answer:RNS, modified throughout the text
- Line 142: „dyosmin” or rather „diosmin”:
Answer:diosmin, modified
10. Line 171: „The IM depends on the host, provides benefits for same [16], and…. – unclear: Answer:The grammar was revised and the idea was modified.
- Line 367: Puerarin – please give examples of the plant occurrence of this compound:
Answer:Puerarin is obtained from the plant Pueraria montana var. lobata is a subspecies of Pueraria montana is a natural flavonoid that reduces the production of ROS, and renovates the antioxidant enzymatic system, regulates the expression of the S genes, of biosynthesis and of lipid metabolism in liver; it is used in soups, creams, stews, fillings, sauces or jams
- Line 356: „Liver”, maybe better „Liver disorders/ diseases” as in the other subsections, diseases are described, not individual organs:
Answer: was modified (Liver diseases)
- Line 448: in which aspect Vitamin E is described, since the paper concerns phytochemicals. The same Line 519: vit C and Line 555: vit D:
Answer:The paragraphs and references about these vitamins were removed from the text
- Line 739: what the authors have in mind: “phytochemicals, and principally their bioactive compounds”? after all, phytochemicals are plant-based bioactive compounds produced by plants:
Answer: as you mention, phytochemicals are plant-based bioactive compounds produced by plants, therefore, the text was modified

Round 2
Reviewer 3 Report
Comments and Suggestions for Authors
The authors responded to the comments made.